# “*At the Root of COVID Grew a More Complicated Situation*”: A Qualitative Analysis of the Guatemalan Gender-Based Violence Prevention and Response System during the COVID-19 Pandemic

**DOI:** 10.3390/ijerph191710998

**Published:** 2022-09-02

**Authors:** Luissa Vahedi, Ilana Seff, Deidi Olaya Rodriguez, Samantha McNelly, Ana Isabel Interiano Perez, Dorcas Erskine, Catherine Poulton, Lindsay Stark

**Affiliations:** 1Brown School of Social Work, Washington University in St. Louis 1 Brookings Drive, St. Louis, MO 63130, USA; 2UNICEF Guatemala, 13 Calle 8-44, Cdad. de Guatemala 01010, Guatemala; 3UNICEF Headquarters, United Nations Plaza, New York, NY 10017, USA

**Keywords:** Guatemala, gender-based violence, GBV prevention and response, gender mainstreaming, COVID-19, LMIC

## Abstract

A growing body of literature has documented an increased risk of gender-based violence (GBV) within the context of COVID-19 and service providers’ reduced capacity to address this vulnerability. Less examined are the system-level impacts of the pandemic on the GBV sector in low- and middle-income countries. Drawing on the perspectives of 18 service providers working across various GBV-related sectors in Guatemala, we explored how the Guatemalan GBV prevention and response system operated during the COVID-19 pandemic. Findings highlight that the pandemic reinforced survivors’ existing adversities (inadequate transportation access, food insecurity, digital divides), which subsequently reduced access to reporting, justice, and support. Consequently, the GBV prevention and response system had to absorb the responsibility of securing survivors’ essential social determinants of health, further limiting already inflexible budgets. The pandemic also imposed new challenges, such as service gridlocks, that negatively affected survivors’ system navigation and impaired service providers’ abilities to efficiently receive reports and mobilize harm reduction and prevention programming. The findings underscore the systemic challenges faced by GBV service providers and the need to incorporate gender mainstreaming across public service sectors—namely, transportation and information/communication—to improve lifesaving GBV service delivery for Guatemalan survivors, particularly survivors in rural/remote regions.

## 1. Introduction

COVID-19-related movement restrictions and adverse socioeconomic impacts have exacerbated the “shadow pandemic” of gender-based violence (GBV) [1,2]. Numerous publications have underscored how the pandemic has contributed to substantially increased incidence and severity of GBV globally [2,3,4,5,6,7,8], mimicking patterns observed during previous infectious disease epidemics, such as Zika and Ebola [9,10,11]. These increased experiences of violence can largely be attributed to existing gender-insensitive policymaking systems that do not address structural and environmental risk factors for violence [10,11,12,13]. The availability and accessibility of GBV services have also shifted during the pandemic: justice/legal support systems and emergency shelters for GBV survivors have been particularly impacted, limiting survivors’ access to critical safety services and increasing time between disclosure and support [6,7,8].

The GBV prevention and response system is the medium through which GBV survivors and service providers interact to facilitate the reporting of violence and/or access to care. Scholarship has detailed how isolation in the home, unemployment, and movement restrictions have magnified GBV risk [1,14,15,16], as well as highlighted the resource and burnout challenges faced by service providers trying to address this increased vulnerability to violence [5,17,18]. Such literature has advanced the understanding of how and why the COVID-19 pandemic has magnified individual-level risk of violence and/or reduced the capacity of service providers to meet this increased vulnerability.

Less investigated has been the manner in which GBV prevention and response systems operate as a whole within the wider context of COVID-19. Adopting a complex systems science theoretical position [19,20,21] provides a useful lens to better understand how the individual-level risks, behaviors, attitudes, and challenges faced by survivors and service providers are reinforced by system-level structures and policies, which are in turn responding to the COVID-19 crisis. Moreover, identifying leverage points where the GBV prevention and response system is failing to meet the needs of survivors and service providers can support systemic change.

The interconnected sectors comprising the GBV prevention and response system are acted upon exogenously by the social and political environment. The COVID-19 pandemic can be conceptualized as an exogenous shock to existing GBV systems. Previous research has shown how this shock has magnified GBV, altered reporting, and increased vulnerability to GBV risk factors in high, low-, and middle-income countries (LMICs), as well as fragile states [16,22]. However, within the context of LMICs, less is known about how the pandemic context has altered the functioning of GBV systems, intersectoral coordination of services, and the provision of quality digitized services [17,23,24]. Gender-sensitive systems, perspectives, and approaches to pandemic control have been extremely limited, thereby inhibiting effective GBV prevention and response during the pandemic, particularly in LMICs and Latin America [25].

Guatemala, though classified as an upper-middle income country, consistently ranks among the top nations in the world for malnutrition, and experiences significant inequality and persistent poverty [26]. Guatemala is part of a significant regional trend in migration: while an estimated 378,000 people annually leave Central American states and journey north toward the United States [27], Guatemala itself is also home to over 180,000 refugees and asylum seekers [28]. Numerous forces drive migrants north from and through Guatemala, including poverty, government corruption, organized crime, and high rates of domestic violence [29]. Despite a sweeping 2008 law overhauling the response to GBV and femicide in Guatemala, 600–700 women are still murdered annually [30], and the lifetime prevalence of intimate partner violence (IPV) in Guatemala was estimated to be 21% in 2018 [31]. Although the 2008 national legislation aimed to decrease the incidence of GBV and increase support services for survivors, service availability is still insufficient throughout the country, particularly for women living in rural areas [32].

In Guatemala, services for protection from and prevention of GBV are covered by government institutions and nongovernmental organizations (NGOs) that refer cases of violence to the Public Ministry (autonomous institution) or National Civil Police (government body) for criminal investigation and criminal prosecution. The entity in charge of supervising that all institutions function properly is the Human Rights Ombudsman. The Guatemalan GBV prevention and response system is largely decentralized, meaning that few service providers are able to respond to violence cases across the country. There is a lack of coverage in rural and remote areas and among Indigenous communities, who require linguistic inclusiveness.

In March of 2020, Guatemala instituted what was initially proposed as an 8-day national lockdown to attempt to control the spread of COVID-19, which continued through October 2020 due to continued peaks of the virus [33,34]. The pandemic also contributed to increased poverty, bringing poverty levels to 47% by the end of 2020 [26]. As in many corners of the world, GBV in Guatemala worsened during the pandemic, with cases of violence spiking in direct correlation with the onset of the national lockdown in late March 2020 [34].

By employing a systems science perspective to conceptualize the Guatemalan GBV prevention and response system, this research qualitatively explored how the COVID-19 pandemic has affected the Guatemalan GBV sector. Drawing on 16 in-depth, semistructured interviews conducted with GBV service providers working in the Guatemalan GBV prevention and response system, we explored how the COVID-19 pandemic has impacted not only individual women, girls, and service providers but also how the pandemic has impacted the broader GBV system and its interrelated sectors.

## 2. Methods

The present analysis, focused specifically on Guatemala, is part of a multicountry project conducted in LMICs that explores the impact of the COVID-19 pandemic on women and girls’ safety, their knowledge of and access to GBV services, and adaptations to service provision in light of COVID control measures [35].

### 2.1. Data Collection

To understand how GBV service providers operated within the context of COVID-19 and to gain insights into survivors’ experiences in navigating the system, we interviewed various individuals working within the Guatemalan GBV prevention and response system as key informants. Key informant interviews are semistructured interviews commonly used in public health research, wherein participants are probed to share their expert perspectives and reflections on the community [36]. A semistructured key informant interview (KII) guide was developed by the research team in partnership with GBV personnel from the in-country UNICEF office. Questions covered a range of topics, including previewed changes in the experiences of violence among women and girls during the pandemic, changes in service provision, experiences with virtual service provision, system coordination, challenges faced by providers, and provider mental health, among others. Purposive sampling was used to identify participants that could serve as key informants (KI). To purposefully collect a sample of experienced GBV service providers working across multiple GBV prevention and response sectors, UNICEF partners compiled a list of service providers from both government and NGOs across a range of service sectors, including but not limited to antiviolence centers, the justice sector, shelters, hotlines, and antitrafficking organizations. Service providers were initially invited to participate in the study through a recruitment email that explained the purpose of the study. Individuals who expressed interest in participating were contacted by a member of the research team to schedule the interview. Twenty-six service providers were invited to participate, 17 individuals followed through to schedule interviews, and 16 providers attended their interviews. Additionally, one service provider brought two additional colleagues to the interview, for a total of 18 KIs across 16 interviews.

All interviews were conducted virtually using the Zoom platform. Prior to the start of the interview, the researcher reviewed the consent form and allowed the participant to ask clarifying questions. Verbal informed consent was obtained. All KIIs were conducted in Spanish and lasted approximately 60 to 90 min. All study procedures were approved by the Health Media Lab Institutional Review Board (361GLOB21).

### 2.2. Analysis

KIIs were audio-recorded, transcribed, and translated into English. All names, locations, and other identifying information were removed from the transcripts to ensure the confidentiality and anonymity of study participants. The team used a combined deductive and inductive approach to refine a codebook previously employed in a multisite study [35,37]. After reading through the transcripts and memoing, a team of four researchers iteratively revised the codebook by adding data-driven codes based on emerging themes from the Guatemalan data. Following the creation of a draft codebook, the team co-coded a subset of transcripts to ensure interrater reliability before coding the full data set. Coded excerpts were then organized in a data display to thematically understand how COVID-19 impacted the GBV prevention and response system and the interrelated sectors. All analyses were conducted in Dedoose [38].

## 3. Results

### 3.1. Sample Description

The research team interviewed 18 KIs from governmental (*n* = 13) and nongovernmental (*n* = 5) organizations in Guatemala, including antiviolence organizations (*n* = 9), branches of the justice system (*n* = 7), and crisis and support hotlines (*n* = 2). The antiviolence organizations included political advocacy and anti–human trafficking groups, as well as shelters and child-protective services. The informants from the justice system represented governmental bodies that are in charge of receiving and investigating GBV cases, providing legal representation to survivors, and prosecuting GBV cases. Half of the KIs (*n* = 9) were part of organizations that primarily offered services to adult survivors, while the other half (*n* = 9) focused on minors.

### 3.2. Themes

The COVID-19 pandemic and corresponding control measures had a twofold effect on the Guatemalan GBV prevention and response system. First, the pandemic reinforced survivors’ existing adversities, which subsequently reduced access to reporting, justice, and support. Consequently, the GBV prevention and response system had to absorb the responsibility of securing survivors’ essential social determinants of health (SDH). Second, the pandemic imposed new challenges that negatively affected survivors’ navigation of the system and impaired service providers’ ability to receive reports and mobilize prevention/response. Based on service providers’ reflections, we explored the Guatemalan GBV prevention and response sector within the COVID-19 context through five interrelated themes: (1) addressing immanent SDH, (2) altered patterns of reporting, (3) resource challenges, (4) digitizing service provision, and (5) navigating health–legal–social services.

### 3.3. Addressing Immanent SDH

Multiple service providers across sectors unequivocally expressed that the COVID-19 pandemic magnified and reinforced the backdrop of adversity faced by GBV survivors. Considering the absence of robust gender protections, service providers expressed that the pandemic worsened several mutually reinforcing SDH. Specifically, precarious employment, inadequate water and sanitation infrastructure, scarce electricity, malnutrition and food insecurity, and limited public transportation were mentioned. Simply stated by a KI working for the justice system, “*basically the needs [of victims] are the same, but they’ve increased*” (KII_16). Moreover*,* “*if a victim was economically dependent before, now even more so*” (KII_16).

Further, “*victims in rural areas lack access to electricity, and obviously to the internet or a phone line. Poverty and extreme poverty are also obstacles that are exacerbated by the pandemic*” (KII_16). Remote regions in Guatemala were described as “hard to reach” in terms of service access because the communities are “*eight or nine hours away*” and service providers had to “schedule interventions in advance” (KII_15). Discrimination and limited resources in remote areas interacted with the pandemic to further marginalize rural and Indigenous populations:

*“The large inequalities that exist are also related to discrimination and the lack of access [to resources]. If we’re talking about rural areas and urban areas, this is also something that has a lot to do with discrimination relating to ethnic groups, our Indigenous population”* (KII_18).

Service providers highlighted that pandemic-related unemployment had a particularly profound effect on women and girls who mostly relied on informal and gendered forms of employment, such as domestic work and care labor. Gendered economic precarity and food insecurity, incubated within a context of unemployment, underemployment, and inadequate social protections, were described as being particularly striking for sex workers and transgender women. As noted by one service provider, “*I know that [organizations that work with trans women] distributed a lot, a lot, a lot of bags with food last year, particularly for trans women who are sex workers and were not able to work during the first months of the pandemic*” (KII_12).

Service providers working in the child-protection sector reflected on how the pandemic affected child and household malnutrition and food insecurity. Consequently, the governmental child-protection and foster-care sector had to shift their focus from providing specialized services to providing humanitarian aid:

*Severe child malnutrition increased because … many people and families lost their jobs, they were fired, and this obliged us, as an institution, to coordinate some humanitarian aid, the inclusion of some social programs, but these were very limited, very limited considering the enormous need that was just identified and which was getting worse, it was insufficient, any aid sent was insufficient against the needs, and we don’t have a welfare approach, we are rather focused on providing special care, on being able to provide psychosocial, pedagogical, legal care to restitute the rights of children.* (KII_5)

Some service providers also mentioned substandard water and sanitation infrastructure, which represents an unmet basic human need that escalated in priority during the pandemic. For example, a service provider working at a GBV crisis and support line noted:

*During the pandemic we had a lot of complaints about access to drinking water. There was a lot of emphasis about handwashing, but many sectors lack access to drinking water, so the number of complaints about this skyrocketed. In the case of water-related interventions, we labeled them as emergencies, as health-related. The idea is to approach the responsible institution and get a timely response in order to reestablish the people’s human rights.* (KII_4)

Service providers also mentioned that pandemic-related economic losses increased financial power differentials between abusers and survivors, making it more difficult for survivors to flee abuse. For instance, from the perspective of a KI working for an NGO serving the Indigenous Mayan community in northwestern Guatemala:

*The economic dependence of women also forces them to endure violent situations. It is frequent, for example, to hear, ‘Look, the thing is, if I leave my husband, who’s going to support my children?’ At the root of COVID-19 grew a more complicated situation.* (KII_6)

The same service provider revealed that in an effort to mitigate pandemic-related food insecurity, the NGO supported women to develop family gardens in the community. She explained that “*in some cases, there has been an opportunity to be able to sell [products], and also to generate some economic income.*” Even families without land were supported to cultivate “*inclusive small-scale gardens*” in pots (KII_6).

Another service provider noted that household economic precarity magnified during the pandemic and contributed directly to violence against women and children.

*“Physical abuse increased because of confinement and the economic crisis. Economic challenges always tend to create tensions, and now that everyone was forced to stay in the house for a long time, these tensions grew even further. In many cases, we saw that the husband would abuse his wife, and then she would abuse her children. Violence became commonplace, a daily event even.”* (KII_2)

Lastly, access to affordable public transportation emerged as a key social determinant of health among Guatemalan GBV survivors. As mentioned in an unprobed manner by several KIs, pandemic-related movement restrictions severely limited access to public transportation and resulted in price surges. Weakened public transportation systems negatively affected survivors’ ability to leave abusers and access reporting mechanisms and services. A service provider working with minors in the criminal justice system noted that when the pandemic started, “*the first thing done was to restrict several rights, mobilization among them one of the most particular things restricted was public transportation services. So, that restriction, in particular, means that the population does not have easy access to practically anything*” (KII_9). Service providers’ abilities to mobilize antiviolence programs and initiatives was also impacted by reduced access to public transportation:

*Teams still can’t access all [regions] … Many of them have mobilized themselves by bus, public transportation, and some even have their own vehicles, or are paid an allowance through transportation lists. However, right now, under the pandemic, there is no access to regular transportation, and when transportation started to reactivate, the cost was double and even triple, which meant that budgets were affected. In the end, if there is no tool to bring the services close to communities, it is a challenge for us to be able to solve it. These are the most significant challenges we face* (KII_5).

### 3.4. Perceived Patterns of GBV Reporting and Processing

KIs mentioned that GBV reporting during the pandemic was affected by a variety of factors, including movement restrictions, human resource challenges, prioritization of infection control over violence protection, reporting mechanism adaptations, and reduced public transportation. Multiple KIs also noted that when system-level gridlocks decreased reporting and access to justice, cases were less likely to move forward, thereby making it harder to hold perpetrators accountable. Therefore, service providers worried that decreases in reporting reinforced perpetrator impunity. As noted by one key informant, “*the pandemic increased violence while also reducing the possibility for victims to access justice*” (KII_16). Providers believed that reduced access to formal reporting mechanisms was at least partially due to “*the many roles that a woman now has to fulfill: they are mothers, sisters, grandmothers, caretakers, or educators of their children at home.*” (KII_16).

Other service providers offering legal and psychological services to survivors noted that movement restrictions and stay-at-home orders reduced formal reporting: “*Confinement forced people to stay inside and stop reporting abuse … They didn’t have any mechanisms or means to escape or file a report*” (KII_1). Service providers also expressed that reporting patterns were affected by pandemic-related limits to public transportation. For instance, when asked what groups of women could not access services, a service provider working for an NGO serving child and adolescent survivors of sexual violence stated:

From March to April [of 2020] women who didn’t own a car couldn’t even file a report. The phone line for filing reports hadn’t been implemented yet. Public transport services resumed in September. There was also a curfew. From May onwards, we could go out from 6:00 AM to 2:00 PM, and then the curfew would start. So, we had very little time. We could only open for 4 h, so only people with a car could commute to file a report. Those who didn’t own a car were the most affected. (KII_2)

Service providers noted that due to movement restrictions, such as curfews and lockdowns, survivors were not able to report violence or seek help in a timely manner. Consequently, case severity and lethality increased, leading to fatalities. One service provider explained:

*“Yes, [most cases] were more severe compared to before the pandemic. Women disappeared before the curfew began. Since their family couldn’t get out to activate the alert, these alerts would be sent until later. There was a case in which the neighbors of a missing woman heard her little girl crying. The woman was found dead two days later… [The perpetrators] took advantage of the curfews to get rid of the bodies.”* (KII_2)

Further, service providers revealed that a reduction in formal reporting led to perpetrator impunity. For instance, from the perspective of a KI working to monitor and coordinate medical care of sexual violence survivors, “*We have the same [violence] figures we had in previous years, the only difference is that there will be more impunity now because people couldn’t go out to file a complaint. That is why we also encourage people to report and to stop normalizing sexual violence.*” (KII_14) The reduced ability of antiviolence organizations to mobilize community outreach focused on GBV awareness and reporting contributed to a culture of violence normalization, which in turn further reduced reporting:

*The problem was that lockdown didn’t allow us to deploy people or to organize any mass activities, we couldn’t organize meetings with parents, children or institutions. We had to change the strategies and find ways for the people to stay informed and specially to be made aware because sexual violence, exploitation and people trafficking are crimes that most people have normalized, so there isn’t much of a culture of reporting these crimes, sometimes this happens due to a lack of credibility and of course this creates impunity* (KII_14).

The impunity of GBV perpetrators was also described as occurring in part due to the concentration of law-enforcement efforts on enforcing pandemic control policies. Thus, even when violence was reported/disclosed, the capacity of law enforcement and the judicial system to hold perpetrators accountable was more limited during a pandemic context:

*Full lockdown caused an increase of human rights violations and a lack of response from the system. There was an increase of missing girls but a reduction of reports in the system. When we started looking into what was happening, we realized that the police were no longer accepting any reports during lockdown so if people went to a police station to report a missing girl or woman, they said that they were asked to focus on the pandemic and that they couldn’t deal with other types of reports* (KII_3).

In contrast, one KI working for the justice system who oversaw the investigation of specialized GBV cases noted that the increased police presence intended to enforce pandemic control measures was helpful for more quickly responding to GBV:

*“Because there are lockdowns and there’s a need to verify that people are staying home during lockdowns, the police became present throughout the whole country, that is, they monitored the streets. Therefore, when there was a violent act, the police were nearby and could aid the victims. So, we do believe that the Public Ministry and the National Civilian Police have helped”* (KII_16).

Importantly, a few KIs did not feel that GBV reporting decreased during the pandemic. One service provider reported that, in her experience, when multiple avenues to report violence were made available, reporting increased:

*I think the increase in complaints is a positive thing. It means women feel more empowered to report and they have more trust in the system protecting their rights … In terms of technology, I think that was great progress too. We were able to get a lot of complaints that way. People just have to tag the Procuraduría de los Derechos Humanos [Attorney General’s Office for Human Rights: the governmental body seeking to secure constitutional human rights], and the consultant is monitoring that. They get their information, like a phone number, and they give that to us. And then the officers here can follow up* (KII_4).

However, while another service provider also noted an increase in crisis and support-line calls, they were often unrelated to GBV, and as a result, the KI encountered challenges in identifying and responding to GBV cases, given the fixed budget and staff:

*The pandemic has been a challenge for us and for 1555 [the number for a national hotline that receives complaints, including GBV], because initially we received an overflow of requests. Not all were necessarily related to human rights violations, but there was a lot of uncertainty about the disease, healthcare services, and a large number of cases. People, because they trust us, came to us* (KII_4).

### 3.5. Providers’ Research Challenges

Inadequate and inflexible budgets, staffing limitations, and personal protective equipment (PPE) constraints comprised some of the resource challenges faced by KIs during the pandemic. Service providers explained that multiple sectors could not expand their budgets or be flexible in allocating additional expenses within the COVID-19 context. Simply put, “*if something is not in the budget, we cannot get it*” (KII_14). For example, KIs conducting specialized investigations of serious GBV cases recognized that existing budgets could not accommodate additional costs associated with purchasing PPE and gasoline for private transportation:

*“The pandemic struck the world in March 2020, when there was already an annual operating plan created based on another plan conceived for normal times. Some of the resources had to be used for health security equipment to prevent the institution’s staff from getting infected. Therefore, there was a shortage of other things needed, for example—and this is essential—gasoline for vehicles. By using part of the budget for health security measures, other services lose resources. And one of the resources that has been severely affected was gasoline for transporting prosecutors, victims, in the institution’s vehicles”* (KII_16).

The shift to virtual service provision further limited budgets. As stated by one KI: “*Working in a digital environment also requires funds because the internet isn’t free. All of this has modified the budget.*” (KII_16). As digitized services were not originally planned, organizations suddenly found themselves with unexpected expenses, such as professional Zoom accounts: “*At the beginning we hadn’t considered that, Zoom gave you 40 min. but it wasn’t enough, and we had to pay for that*” (KII_14). Further, a service provider in the child-protection sector noted that resource constraints during the pandemic impacted their ability to respond to child maltreatment cases in a timely manner:

*“We don’t have ideal resources. With the few investigators, the vehicles that we have, sometimes we are not able to investigate all the cases at once, and the cases start accumulating. And this harms children because if there is a case of maltreatment right now and I don’t have resources, I don’t go now. I go next week. So that puts children’s lives at greater risk. Yes, it has had an impact on us because, as there have been many complaints, our work, I would say, was not carried out immediately”* (KII_11).

Lack of access to PPE was consistently referenced as a significant challenge, given that, “*suppliers were out of stock in every sense, biosecurity was the most complicated issue*” (KII_7). When staff tested positive for COVID-19, the GBV response system faced human resource shortages, thereby resulting in slowed or halted service provision. A KI in the justice sector stated there was a reported “*decline in the prosecution services*” *given that* “*staff started being ill and they had to [delay] all procedures*” (KII_4). Physical distancing and lack of PPE also inhibited the capacity of the attorney general’s office to investigate GBV cases. For instance, one KI stated that they could not “*carry out the same number of investigations*” because:

*“We had to use [PPE], which wasn’t always available for the interventions… Access to victims was also limited for the same reasons. Like, ‘look, this might be a victim, but they’re in a restricted area of the hospital, so you can’t access to interview them.’ This also led us to reinvent ourselves. We said, ‘we’ll make a phone call; if you won’t let me in, take this phone to the person so I can talk to them.’ And this deals with access to technology, which institutionally is limited. So, the investigators had to use their own means and resources to investigate”* (KII_4).

In addition to acknowledging PPE inaccessibility, multiple service providers indicated that GBV service providers had not been prioritized for COVID-19 vaccinations, despite the essential nature of their work:

*“It has been stressful, very stressful to know that the fact that we also have to expose ourselves to work, to go out, to presential meetings, have us in a greater level of exposure, and that as a country we still do not have enough vaccines to ensure that we, from the justice system, have access to this option. But we have had enough colleagues who have died or who have fallen seriously ill”* (KII_9).

Service providers expressed general concern regarding slow vaccine distribution, saying, “*Despite the fact that the vaccine is already available, the vaccine rollout hasn’t been as quick as expected. Only a small percentage of the population has been vaccinated*” (KII_2). Service providers also voiced concerns regarding health-system governance that may have negatively interfered with vaccine distribution:

*“Another thing I think of is that the whole issue of corruption in the country … The health system is completely collapsed, but it’s collapsed because the money has been stolen! I mean, what do you do when there are no supplies? I mean, right now they [the government] … bought a lot of vaccines but it is uncertain when they will arrive, and the State is questioned and the State doesn’t know when the vaccines will be sent”* (KII_12).

Despite the myriad resource challenges faced by providers throughout the pandemic, several KIs noted that a few silver linings emerged. One service provider from the justice sector believed that the hardships helped her colleagues learn how to “*take care of ourselves*” and “*coordinate better.*” As a result, they said, “*the institutional coordination improved a lot, fellowship improved a lot*” (KI_11). Interviews also highlighted providers’ resourcefulness in linking women to services, despite a lack of resources. For example, in order to ensure service continuity for rural women in need of remote mental health and psychosocial support, providers would “*top up [a client’s] phone so that she could receive therapy*” (KII_2). Additionally, the KI described an innovative partnership wherein Uber provided promotional codes for survivors to attend court hearings when access to transportation was lacking (KII_2).

### 3.6. Digitizing Service Provision

Participants discussed challenges encountered by survivors and service providers when digitizing or attempting to digitize service provision or reporting. One of the most frequently discussed challenges was digital divides (DDs): inadequate access to information and communication technology. Participants mentioned that DDs intersected with existing gender inequities, poverty, and urban/rural disparities, and that the COVID-19 pandemic further marginalized certain women and girls from accessing technology and, therefore, GBV services.

As noted by one key informant:

*There’s a lack of culture of accessing public information. Women lack access to mass media. And this is on top of the victims’ lack of knowledge about services provided through platforms and phone calls. So, all these situations add up* (KII_16).

Poverty was also described as exacerbating the DD. Among survivors living in poverty, the need to secure imminent survival needs outweighed the potential benefits associated with securing the technological prerequisites for GBV care. A KI working to provide free legal and psychological services to survivors described the trade-off between survival and technology needs:

*“When there are very high poverty levels, what humans do is try to find survival mechanisms, and what are our basic needs? Food and shelter. These are our main concerns because they allow us to survive. I can live without Internet reception or without a phone, but I cannot survive without any food or without a place to live”* (KII_1).

Limited access to technology also directly impacted Guatemala’s pandemic response. DDs set the stage for inequitable vaccine access: “*From the government, even right now, it’s being said for those who want to get vaccinated, they must register using a smartphone. Who has a smartphone, internet service, and the ability to drive, and to enroll?*” (KII_6).

Service providers also highlighted that DDs widened urban–rural disparities concerning GBV service accessibility. Rural areas were described as having weaker internet and electrical infrastructure:

*“Victims in rural areas lack access to electricity, and obviously to the internet or a phone line... Even in super urban areas, there is an Internet problem. Even more so in rural communities. And that should not be forgotten in the case of Guatemala... So then what’s proposed to be done in virtually in some cases isn’t possible everywhere”* (KII_16).

On account of gender inequities, poverty, and urban/rural disparities, DDs were key barriers to receiving online/virtual services:

*“Even if there is the possibility to file reports and submit your documents online, most people still go to the bricks-and-mortar offices because they have no internet access. Sometimes, they own a smartphone but have no credit or anything. They sometimes only top up their phones with 5 quetzales that day to be able to send an email, but they can’t afford to spend more than that to take their classes or receive therapy. They were forced to prioritize their resources”* (KII_2).

Another service provider working with the Indigenous Mayan community reflected on the limited technological resources available to GBV prevention and response organizations as well: “*The Women’s Prosecutor’s Office deals directly with victims a lot. And we certainly can’t communicate virtually with a victim because of their resources, and also because of the institution’s resources. Internet access is limited. It all goes back to the budget*” (KII_16). Further, “*prosecutors didn’t have computers with cameras for the virtual hearings,*” thereby further contributing to justice system delays (KII_16).

### 3.7. Navigating Health–Legal–Social Services

Service providers expressed that even when GBV was formally reported, survivors’ ability to navigate the health, legal, and social sectors involved was circumscribed by halted legal proceedings, diminished intersectoral collaboration, and the reduced capacity of emergency shelters. Regarding the justice sector, multiple service providers indicated that at the beginning of the COVID-19 pandemic, “*all the courts closed … due to containment*” and “*judicial processes of [cases] stopped*” (KII_7). These suspensions generated a backlog in the justice system, wherein “*some cases will even take 4 years to get to trial due to these delays and the number of reports filed last year [2020]”* (KII_2).

In response to court closures, service providers indicated that women and girls with open cases awaiting trial “*were taken aback*” and subsequently asked, “*When are we going to get to trial? Will my case even get to trial? When will hearings resume?*” (KII_2). In response to survivors’ concerns, a service provider aiding survivors seeking justice and protection stated, “*We tried to calm the users down so that they didn’t panic and felt like they wouldn’t get the outcome they expected*” (KII_2). The same service provider stated:

*“We were not ready for such a crisis, no one was, so we didn’t know how we could coordinate rescue efforts with the government. We had to learn this on the go. We would call the Prosecutor’s Office and the user from our home. It was very complicated, and on some occasions, we could have handled things better, but even the government didn’t know what to do”* (KII_2).

Existing projects and initiatives planned by NGOs also had to be halted due to pandemic-related constraints. Thus, given fewer available community programs, survivors had fewer opportunities to connect with the GBV system:

*When we were finally able to provide our services on a normal basis, we had to focus on guaranteeing the safety of everyone, both our staff and the people we serve…The projects we were involved in the past were mostly field projects, and they had to stop, so later we had to see how we could resume them while performing other tasks at the same time. This has been an exhausting year for sure* (KII_2).

Pandemic-related movement restrictions also negatively affected admittance to emergency shelters. One service provider highlighted that survivors could not be admitted to shelters, thereby forcing survivors to return to the abuse, potentially leading to increased lethality:

*Most of the shelters were closed, so women had nowhere to go, so that meant they had to stay at home. In those cases, our suggestion was to get the perpetrator out of the house. Authorities started doing this because there were shelters that weren’t able to relocate the people they had taken in. Shelters had to figure out a way to provide food for all of them … These shelters are closed spaces, so you need to guarantee people’s safety* (KII_2).

Further, pandemic-related curfews dictated the specific times when survivors could be admitted to shelters:

*We didn’t have permission to go out after the curfew. Only doctors, prosecutors, and the police could. Also, most of the admissions needed to happen after the curfew, at 5:00 PM or 8:00 PM, so even if we had everything ready, if the person couldn’t get to the shelter before the curfew, then she wouldn’t be admitted* (KII_2).

Despite technological limitations, one service provider described the innovative strategy of partnering with law enforcement to bolster antiviolence campaign messages:

*We noticed that the police cars of the [Civil National Police] had loudspeakers and so we realized that our consultants who had been deployed to the departments could coordinate and move around the areas even during non-working hours, after the curfew, and deliver our messages. That was one of our strategies. The campaign we spread through social media and the radio... We implemented a strategy so that our team could approach the authorities, and we implemented ways of delivering information virtually* (KII_14).

While the pandemic negatively affected coordination between different sectors involved in GBV prevention and response, this was not uniformly experienced by all service providers. One KI working for the justice system noted that the pandemic prompted the Public Ministry to establish prosecutor’s offices in every municipality in the country, meaning that “*victims don’t have to travel to the headquarters of their region or to the capital*” to file a report or request security measures (KII_16).

## 4. Discussion

Drawing on the perspectives of 18 Guatemalan GBV service providers, this research qualitatively explored how the COVID-19 pandemic affected the Guatemalan GBV prevention and response system and related interactions between survivors and service providers. Our analysis highlights the interplay between sectors within GBV prevention and response (medical, legal, mental health, survivors’ assistance) and between the GBV system and other sectors, such as transportation and information/communication. Ongoing state fragility and lack of COVID-19 vaccine availability in Guatemala further eroded GBV system strengthening and gender mainstreaming efforts.

The COVID-19 pandemic and associated control measures fostered an environment in which poverty, food insecurity, unemployment, lack of electricity, precarious labor, and weak transportation infrastructure became increasingly prominent challenges for Guatemalans. The already high prevalence of food insecurity and malnutrition in Guatemala [39,40] was heightened by the increasing economic precarity engendered by the pandemic. This pandemic-driven deterioration of household economic and nutritional resources is also reflected in other LMICs [41].

Prior to the arrival of COVID-19 in Guatemala, 80–90% of the urban poor and Indigenous women [40,42] and more than 70% of the overall population worked in the informal sector [43]. The International Labour Organization estimates that informal workers saw their earnings decline by 60% in the first months of the pandemic, underscoring the dire financial situation in which many Guatemalans found themselves [44]. In response to deteriorating SDH, many GBV service providers mentioned being compelled to shift or expand services to meet clients’ basic needs, a pattern that has been similarly noted in high-income countries [17,45]. In our data, several service providers alluded to the financial and resource challenges posed by the pandemic-related exacerbation of poverty. However, the obligation service providers felt to their clients coupled with the reality that material necessities like food and income may be prerequisites to accessing GBV care motivated adaptation of tightly constrained budgets to provide for SDH. This expansion in the scope of GBV service provision likely detracts resources from providing specialized care or prevention outreach. Furthermore, service providers were required to rapidly build new models of service delivery, adding additional stress and compounding workloads. Pandemic-related service changes raise questions of whether and when agencies will be able to fully return to their GBV mission-driven services and if critical GBV services and referral networks have maintained quality.

COVID-19 precipitated significant gridlocks and delays in service access. The justice sector in particular experienced prominent delays above and beyond prepandemic access difficulties [32,46]. Similar pandemic-induced delays and gridlocks in law-enforcement and justice sector services have been noted in other LMICs [41]. As survivors experienced trial delays during the pandemic, service providers underscored the perceived culture of impunity concerning GBV perpetrators. When survivors are forced to decide whether to report abuse and pursue legal action against abusers, they undergo a nuanced decision-making process in which they weigh the costs and benefits of reporting. During the pandemic, the potential costs of reporting grew to include possible COVID-19 infection, navigating movement restrictions, and limited public transportation. If the justice system cannot respond to GBV or responds with significant delays, then the perceived cost of reporting will likely outweigh the perceived benefit. Gridlocks and delays in accessing GBV services have the potential to fundamentally alter survivors’ decision-making calculations.

Further, mechanisms of GBV service delivery were forced to shift abruptly in Guatemala, particularly with the public transportation suspensions that went into effect from March to August 2020 [47]. These limitations on public transportation made it difficult for survivors to report violence or utilize services they would ordinarily have had in-person access to. Some Guatemalans were able to effectively engage with digitized GBV services. However, it has been globally reported that survivors without access to technology are systematically excluded from digitized GBV services [18,41,48]. DDs and limited access to transportation operated in similar ways to reduce reporting and service access. The groups already most vulnerable, namely, rural women, were less likely to have access to information and communication technology and more likely to live further away from GBV-related centers [49,50]. For this reason, digitizing services did not actually reduce disparities in reporting or access, but rather carried existing disparities into a new medium. Consequently, improvements to national transportation and communication infrastructure are critical to addressing the needs of GBV survivors and strengthening the public sector’s ability to equitably adapt in the face of emerging population health threats. Additionally, these challenges point to the importance of gender mainstreaming and intersectoral collaboration among GBV prevention/response, public transportation, and the information/communication sector: these sectors are interdependent in directly and indirectly moderating access to critical GBV services. Companies like Zoom, Uber, or Lyft might consider offering subsidies to GBV prevention/response organizations as part of their corporate responsibility.

Access obstacles, which were particularly magnified for survivors in rural and Indigenous communities, are rooted in the high concentration of services in Guatemala City and the economic and infrastructure issues that existed in Guatemala prior to the pandemic [51]. Policymakers need to further consider transportation and telecommunications infrastructure strengthening to address GBV response. Furthermore, the economic impact of COVID-19 has demonstrated the need to mainstream GBV prevention and mitigation efforts across sectors [52]. Thus, mainstreaming gender within national-level budgets to address GBV survivors’ barriers to service access should be considered [52]. Governmental organizations in Guatemala also reported limitations due to fixed budgets. Policymakers should consider flexible budgeting to respond to unexpected crises.

Our findings should be considered alongside a few study limitations. First, the findings were drawn from interviews with service providers only. In-depth interviews with female survivors might have allowed for a greater understanding of the challenges they faced in interacting with the GBV system. Second, access to a stable internet connection was required to participate in the KIIs, potentially excluding service providers in rural areas with limited internet access. Future research should consider explicit targeting of providers in rural areas. Given the interviews were conducted in Spanish and the data were analyzed in English, it is possible that some of the nuances of participants’ responses were lost in translation. Finally, this study did not include service providers working for the Ministry of Health’s psychosocial services, who are responsible for providing follow-up care for victims.

## 5. Conclusions

The COVID-19 pandemic and corresponding measures to prevent the spread of the virus have increased the risk of violence for women and girls across the globe and hindered their access to quality GBV-related services. Simultaneously, providers of GBV prevention and response services have been tasked with providing quality care in the face of significant resource challenges, slowed coordination, and minimal protections for their own physical and mental health. Similarly to other studies that have documented challenges in GBV service provision during the COVID-19 pandemic [18,32,40,45,46,47,48,49], findings from this study highlight the inordinate challenges faced by providers (lack of funding, PPE shortages, limited technological infrastructure, and poor cross-sectoral communication and collaboration) and the exacerbation of poverty and adversity faced by survivors. Efforts to strengthen the GBV prevention and response system should include, among other strategies, strengthening partnerships and critical infrastructure for GBV prevention/response, public transportation, and telecommunications, particularly among rural areas and Indigenous populations.

## Data Availability

Not applicable.

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
