# Peer review of "“At the Root of COVID Grew a More Complicated Situation”: A Qualitative Analysis of the Guatemalan Gender-Based Violence Prevention and Response System during the COVID-19 Pandemic"

_ijerph, 2022, doi:10.3390/ijerph191710998_

Round 1
Reviewer 1 Report
The article is an interesting one that studies the corelation between covid and gender-based violence. It successfully provides a detailed information on its subject in the case of Guatemala. The article has the potential of increasing our understanding on a very critical subject from a Latin American perspective/case.
My major comment is as follows:
1. In the conclusion, the article should include a debate how the findings of this article are relevant in the literature. How are this article’s findings relevant/different/similar in comparison to other studies? Are they different providing something we did not know or enhancing what we already know.
2. The conclusion is not satisfactory. The article needs a better, organized and informative conclusion.
3. Similarly, the introduction is a bit unorganized. I suggest a re-perusal of the introduction. It should be more organized in depicting i. the problem, ii. the puzzle, iii. the purpose of the article. Make it more to the point. Avoid indirect samples and examples.
In general, the article has a very strong case-study, but the other parts are not equally successful. This makes the article appearing as a raw stuff. This might be a difficulty for readers who have not direct knowledge on the topic.
I have also some minor comments:
L. 70: LMICS, I guess this as low- and middle-income countries. The text does not provide any information on this.
The introduction should give more information about the GBV Guatemalan prevention and response system. Are they autonomous bodies? Are they reporting the government? Or local authorities? How is the administrative hierarchy? Who works there? How are they financed? Can we get some information about the cultural and religious (as well as educational background) of people who fork for them? A short summary of these services historical background is also needed.
L. 110: Data collection: Though I find the method here consistent, the authors should provide some other works that use same or similar method of data collection. If there is no, a short note on various methods that are used in this vein would be critical. The reader may need to know about the standard in this context.
L 3.2: Sample description: A few words why such a sample is correct to understand the problem. Is this a random group? What is the rationale of putting these people together as sample? The section does not provide the rationale.
Reviewer 2 Report
I appreciate the topics and the significance of the paper. It is well written and the message is clear. However, there is a couple of issues that could be improved or better described.
The authors did not mention how many service providers work in Guatemala (approximately) and if and how well the number of addressed service providers covers the situation in the country.
I would also like to recommend the authors better describe the structure of collected data using KII, especially if there were questions better describing the service providers (who they are [men/women], the services they provide, number of clients they served in questioned period and how is the coverage of the country. Are there regions or counties from Guatemala which were not reported by any service provider or did the group of service providers cover the whole country?
I also recommend adding limitations of study, if there are any.
Author Response
Please see the attachment, under "Reviewer 2."

Round 2
Reviewer 1 Report
I have again read the paper along with the responses. I think the author(s) have made all required updates including adding, changing and revising etc. As it is, I have no further comments or objections. I recommend "publication."